# Comparing Prognosis for *BRCA1*, *BRCA2*, and Non-BRCA Breast Cancer

**DOI:** 10.3390/cancers15235699

**Published:** 2023-12-03

**Authors:** Pedro Antunes Meireles, Sofia Fragoso, Teresa Duarte, Sidónia Santos, Catarina Bexiga, Priscila Nejo, Ana Luís, Beatriz Mira, Isália Miguel, Paula Rodrigues, Fátima Vaz

**Affiliations:** 1Medical Oncology Department, Instituto Português de Oncologia de Lisboa Francisco Gentil, 1099-023 Lisbon, Portugalfvaz@ipolisboa.min-saude.pt (F.V.); 2Molecular Pathobiology Research Unit, Instituto Português de Oncologia de Lisboa Francisco Gentil, 1099-023 Lisbon, Portugal; 3Familial Cancer Clinic Unit, Instituto Português de Oncologia de Lisboa Francisco Gentil, 1099-023 Lisbon, Portugal

**Keywords:** breast cancer, hereditary breast cancer, *BRCA1*, *BRCA2*, risk-reducing surgeries, prognosis

## Abstract

**Simple Summary:**

Approximately 10% of breast cancer (BC) cases are hereditary, and germline pathogenic variants in *BRCA1* and *BRCA2* genes account for 20% of familial BC cases. Long-term follow-up data related to the prognosis and survival of either *BRCA1* or *BRCA2* BC patients are conflicting. The aim of this study is to report the analysis of our cohort of *BRCA1/2* BC patients included in prospective follow-up after genetic testing. We compared clinicopathological characteristics and prognosis between BC patients with *BRCA1* and *BRCA2* and a control group without germline PV (*BRCA-wt*). The presence of *BRCA* mutation confers a higher risk of relapse and death in patients with BC in the Portuguese population. Prophylactic mastectomy and preventive salpingo-oophorectomy confer lower incidence of relapse and longer median invasive disease-free survival and overall survival, respectively.

**Abstract:**

Background: Germline pathogenic variants (PV) in *BRCA1* and *BRCA2* genes, which account for 20% of familial breast cancer (BC) cases, are highly penetrant and are associated with Hereditary Breast/Ovarian Cancer Syndrome. Previous studies, mostly including higher numbers of *BRCA1* BC patients, yielded conflicting results regarding *BRCA1/2* BC outcomes. In the Portuguese population, *BRCA2* BC is diagnosed more frequently than *BRCA1* BC. We aimed to compare clinicopathological characteristics and prognosis between BC patients with *BRCA1* and *BRCA2* mutations and a control group without germline PV (*BRCA-wt*). Furthermore, we explored the frequency and outcomes of risk-reducing surgeries in *BRCA*-mutated patients. Methods: Prospective follow-up was proposed for patients with a diagnosed *BRCA1/2* PV. For this study, a matched control group (by age at diagnosis, by decade, and by stage at diagnosis) included BC patients without germline PV. We compared overall survival (OS) and invasive disease-free survival (iDFS) within the three groups, and the use of risk-reducing surgeries among the *BRCA* cohort. Results: For a mean follow-up time of 113.0 months, *BRCA*-wt patients showed longer time to recurrence (*p* = 0.002) and longer OS (*p* < 0.001). Among patients with *BRCA* mutations, no statistical differences were found, although patients with *BRCA2* BC had longer iDFS and OS. Uptake of risk-reducing surgeries (contralateral prophylactic mastectomy and salpingo-oophorectomy) were negative predictors of invasive disease and death, respectively. Conclusions: Testing positive for a *BRCA* PV is associated with a higher risk of relapse and death in patients with BC in the Portuguese population. Risk-reducing mastectomy and salpingo-oophorectomy were associated with lower incidence of relapse and longer median iDFS and OS, respectively.

## 1. Introduction

Breast cancer (BC) is the most diagnosed malignancy and the leading cause of cancer death in women worldwide, accounting for 11.6% of all new cancer diagnoses and 6.6% of all cancer deaths [1]. Family history is an important risk factor for BC, and circa 15–20% of familial BC can be explained by pathogenic variants (PV) in the *BRCA1* or *BRCA2* genes [2]. Identifying *BRCA1* or *BRCA2* PV has important clinical implications in risk management and cancer treatment decisions. These are highly penetrant genes associated with Hereditary Breast/Ovarian Cancer Syndrome. Women with *BRCA1/2* PV have a lifetime risk of developing BC and ovarian cancer of 45% to 75% and 18% to 40%, respectively [3,4,5]. Besides surveillance with breast magnetic resonance imaging screening, *BRCA1/2* patients may undergo risk-reducing surgeries, including bilateral mastectomy and salpingo-oophorectomy [6,7,8]. The role of risk-reducing hysterectomy is controversial, as some studies suggest a higher risk for uterine cancer in *BRCA*-mutated patients [9,10], whereas other reports attribute a higher risk to tamoxifen use [11].

More than 7000 PV have been identified on these genes, including the Portuguese founder mutation in exon 3 of *BRCA2* (c.156_157insAlu) [12]. This *BRCA2* founder effect contributes to the higher prevalence of *BRCA2* BC in Portugal compared to *BRCA1* BC [13].

Are *BRCA1* or *BRCA2* biomarkers for poor prognosis in BC? Long-term follow-up data related to the prognosis and survival of either *BRCA1* or *BRCA2* BC patients are conflicting. Two large meta-analyses report worse overall survival for both [14,15] when compared to sporadic BC, whereas two other large meta-analyses concluded worse overall survival only for *BRCA1* patients, with similar overall survival for *BRCA2* patients [16,17]. One meta-analysis reports similar survival for both groups [18].

In this study, we report the clinicopathological characteristics of *BRCA1/2* BC patients included in a prospective follow-up registry. Our goal is to compare prognosis-related outcomes—overall survival (OS) and invasive disease-free survival (IDFS)—between BC patients with *BRCA1* and *BRCA2* and a matched control group without germline PV (*BRCA*-wt). We also explore the frequency and outcomes of risk-reducing surgeries in the *BRCA1/2* population.

## 2. Materials and Methods

### 2.1. Patients

A complete personal and family history of cancer is obtained for all patients referred to our multidisciplinary program. Patients are requested to obtain clinical information regarding personal and family cancer diagnosis, especially pathology reports, whenever possible.

Eligibility for genetic testing has evolved within the past decade. Currently, all patients with BC diagnosed under 40 years of age or younger than 65 years of age with triple-negative subtype are invited to undergo genetic testing, even in the absence of family history. Classical criteria for testing include patients with bilateral or multifocal (being one of the neoplasia diagnosed before 50 years of age) or with family history of breast and/or ovarian cancer (one case of each in first-degree relatives, two cases of BC in second-degree relatives younger than 50 years old, or ovarian cancer at any age, or history of male breast cancer). Molecular testing also evolved from sequential testing to multigene panel testing. For diagnosis, these panels include actionable genes [19,20,21].

If considered eligible for *BRCA1/2* testing, patients undergo appropriate counseling and informed consent process. These individuals are invited to participate in a prospective follow-up observational registry, which collects demographic, clinical, and pathological data, as well as data on survival, relapse, uptake of risk-reducing surgeries, and other cancer diagnoses observed during follow-up.

For this study, we extracted data from the registry of female patients with BC as their first cancer diagnosis, diagnosed between January 2000 and December 2022, and with a pathogenic variant in *BRCA1/2* genes.

The control group included BC female patients with BC as their initial cancer diagnosis during the same period but without PV in a panel including *BRCA1, BRCA2, BARD1, CHEK2, ATM, RAD51D, RAD51C, TP53, CDH1, PTEN, BRIP1,* and *PALB2* genes (*BRCA*-wt). The selection of the control group was performed in a 1:1 ratio, matching by age at diagnosis (by decade) and cancer staging.

### 2.2. Statistical Analysis

All categorical variables are presented as numbers and percentages. Given the non-Gaussian distribution of continuous variables, these are presented as median and interquartile range (IQR). Between-group analyses were conducted using the Mann–Whitney U test or the Kruskal–Wallis test, as appropriate, for continuous variables and the χ^2^ test for categorical variables. The Bonferroni correction was applied when necessary.

Inference analyses also included Kaplan–Meier curves and Cox regressions. Dependent variables were invasive disease-free survival (iDFS) and overall survival (OS). For Cox regressions, the backward conditional method was used. Statistical significance was set at *p* = 0.05 (two-sided). IBM SPSS Statistics v23 was used for statistical analyses.

## 3. Results

### 3.1. Patients’ Characteristics

From 5504 cases (4021 index, 1483 family relatives) that consented to genetic testing, 1077 individuals tested positive for *BRCA1/2* PV. A total of 613 were cancer patients, and in 478, BC was the first cancer diagnosis. In 37 patients, BC diagnosis occurred before January 2000, and 88 patients were lost to follow-up. Seven patients were male and, therefore, were excluded from statistical analysis. Three patients had double mutations (*BRCA1* + *CHEK2*, *BRCA2* + *CHEK2*, and *BRCA1* + *BRCA2*). The patient with *BRCA1* + *BRCA2* mutations was excluded from statistical analysis, and for the other two patients, only the *BRCA* PV was considered. Figure 1 shows a flowchart of the patients included in this study.

A total of 684 patients were included in this study (345 *BRCA1/2* and 339 *BRCA1/2*-wt). Among them, 229 (33.5%) were *BRCA2* BC pts, 116 (17.0%) were *BRCA1* BC pts, and 339 (49.6%) were *BRCA*-wt. The median follow-up time was 113.0 months (5.5–281.0). Table 1 includes the patients’ characteristics. *BRCA1/2* patients, as a group, had a median age of BC diagnosis that was non-statistically different from *BRCA*-wt patients. However, *BRCA*-wt patients were older when compared to those with *BRCA2* and *BRCA1* (*p* = 0.020) individually. The distribution of molecular subtypes was different among the three groups of mutations (*p* < 0.001). The number of patients with stage IV BC at diagnosis was not different among the three groups (*p* = 0.357), neither was the relapse rate (*p* = 0.203). Regarding vital status, 24.0% of the deceased patients had a *BRCA2* mutation (*p* < 0.001).

### 3.2. Uptake of Risk-Reducing Surgeries

Regarding *BRCA1* and *BRCA2* patients, 81.3% and 58.1%, respectively, underwent a contralateral risk-reducing mastectomy, a total of 66.8% of all *BRCA* patients. Patients who underwent prophylactic mastectomy had a lower incidence of invasive disease (*p* < 0.001), but no statistical difference was found for iDFS (*p* = 0.543) or OS (*p* = 0.898) (Table 2).

Furthermore, 78.0% of *BRCA1* patients and 76.9% of *BRCA2* patients underwent risk-reducing salpingo-oophorectomy, totalizing 77.3% of all *BRCA* patients. The incidence of invasive disease was found to be higher in these patients; however, they exhibited higher median iDFS (*p* = 0.001) and OS rates (*p* = 0.015) (Table 3).

Recurrence rates were similar between the three groups (23.7% for *BRCA*-wt vs. 21.9% for *BRCA1* vs. 29.3% for *BRCA2*, *p* = 0.203). Locoregional relapse was more common in *BRCA2* BC patients, and distant relapse was more frequent in *BRCA1* BC patients; however, no statistical significance was observed.

During follow-up, subsequent cancers were more frequently diagnosed in *BRCA* patients (19.0% in *BRCA1* vs. 21.0% in *BRCA2* vs. 5.9% in *BRCA*-wt, *p* = 0.002). The most common cancer detected during follow-up was breast cancer for both *BRCA1* and *BRCA2* groups and endometrial cancer for the *BRCA*-wt group. Table 4 further specifies the second primary cancer diagnoses during follow-up.

### 3.3. Invasive Disease-Free Survival and Overall Survival Analysis

For *BRCA*-wt BC, a significantly longer median time to recurrence was observed (95.0 months, 95% CI (72.6–117.4) compared to *BRCA2* patients, 65.0 months, 95% CI (48.1–81.9), and *BRCA1* patients, 31.0 months, 95% CI (13.9–48.1), *p* = 0.002). The Kaplan–Meier curves demonstrate that *BRCA*-wt patients had a higher iDFS than those with *BRCA1* mutation (*p* = 0.002) (Figure 2).

The Kaplan–Meier curves reveal differences in OS among the three groups (*p* < 0.001), with the *BRCA*-wt mutation group of patients exhibiting a higher OS compared to those with *BRCA1* mutation and *BRCA2* mutation (Figure 3).

### 3.4. Multivariate Analysis

We conducted a multivariate analysis using Cox regression, incorporating the variables ‘Mastectomy’, ‘Salpingo-oophorectomy’, ‘Mastectomy + Salpingo-oophorectomy’, ‘*BRCA1* variant’, ‘*BRCA2* variant’, ‘luminal subtype’, and ‘non-luminal subtype’. Our analysis revealed that ‘Salpingo-oophorectomy’ (coefficient: 0.407; *p*-value: 0.006) serves as a negative predictor of iDFS. Conversely, concerning OS, ‘mastectomy’ (coefficient: 0.197; *p*-value < 0.001) emerged as the sole negative predictor. Table 5 describes the results of the multivariate analysis. The complete analysis can be found in Appendix A.

## 4. Discussion

The pattern of *BRCA1* and *BRCA2* variants varies widely among different populations due to the absence of hot spots in *BRCA1/2* genes. For most populations, *BRCA1* PV are either equally or more prevalent in comparison with *BRCA2* PV [22,23,24,25,26]. In some cases, a high frequency of specific mutations has been reported due to founder effects [23,27]. In the Portuguese population, there is a much higher prevalence of *BRCA2* mutations, a phenomenon partially explained by the founder effect of the previously described mutation c.156_157insAlu [12]. In this study, we report the clinical, pathologic, and outcome differences between our cohort of patients with *BRCA1* and *BRCA2* BC included in a prospective registry.

Consistent with previous reports [28,29,30], *BRCA1* BC patients presented at a younger age at diagnosis and with a triple-negative phenotype, whereas *BRCA2* BC was associated with older age at diagnosis and hormone receptor positive phenotype. While the association of *BRCA1* and *BRCA2* PV with breast and ovarian cancer risks is well-defined, the potential association of these variants with other cancers is not so well established. For *BRCA1* patients, studies inconsistently report increased risk of colorectal [31,32], prostate [9,31], and pancreatic [9,32] cancer, as well as cancer of the uterine body and cervix [9], stomach [32], fallopian tube [32], and melanoma [33]. For *BRCA2* patients, there is evidence of increased risk of pancreatic cancer [10,33,34,35], prostate cancer [10,33,34,35,36], and melanoma [10,35]. Reports of excess risk for gallbladder, bile duct [10], stomach [10,34,35], pharyngeal [34], and laryngeal [36] cancers are inconsistent across studies. In our cohort, ovarian cancer was the second most common type of neoplasia detected during follow-up for both *BRCA1* and *BRCA2* patients. The development of second non-breast primary tumors during follow-up was more frequent in *BRCA2* patients, although not statistically significant. The second non-breast primary tumors include a wide spectrum of cancer types, mostly ovary, but also lymphoma, colorectal, pancreatic, and gastric.

The uptake of risk-reducing mastectomy was significantly higher in *BRCA1* patients. However, we did not find a statistical difference regarding risk-reducing salpingo-oophorectomy. We do report higher rates of uptake of risk-reducing surgeries in comparison with other studies. Metcalfe et al. [37] evaluated rates of contralateral prophylactic mastectomy in eight countries (Austria, Canada, France, Israel, Italy, Norway, Poland, and the United States) and reported results ranging between 0.0% and 28.0%, with an average rate of 27.3%. However, large differences by country were evident. Terkelsen et al. [38] evaluated Danish data, reporting a 72% overall rate. Hanley et al. [39] reported uptake of bilateral mastectomy in 55.4% of *BRCA1* BC patients and 58.2% of *BRCA2* BC patients. In our cohort, the contralateral prophylactic mastectomy rate was 66.8% for the overall population. The uptake of preventive salpingo-oophorectomy was 63.2% in the Metcalfe et al. study, 64.7% for *BRCA1* patients, and 62.2% in *BRCA2* patients in the Hanley et al. study, whereas we report 77.3%. This demonstrates the great efficacy of our screening program, as well as a remarkable and close collaboration with surgery departments.

For the described follow-up, iDFS was significantly longer for *BRCA*-wt patients, although our control group belonged to a higher-risk population as patients met the criteria to undergo genetic testing. Differences in OS also benefited *BRCA*-wt patients, with statistical significance. Regarding *BRCA* patients, although we did not find statistical differences in iDFS or OS, *BRCA2* patients had a better prognosis. This is consistent with a predominance of hormone receptor-positive phenotype, known for later recurrence in comparison with HER2 or triple-negative BC [40].

We conducted a multivariate analysis using Cox regression. Our analysis revealed that ‘Salpingo-oophorectomy’ (coefficient 0.407; *p*-value: 0.006) is a significant negative predictor of iDFS. Conversely, concerning OS, ‘mastectomy’ (coefficient: 0.197; *p*-value <0.001) emerged as the sole negative predictor.

The main strengths of our study are the large number of patients, especially regarding *BRCA2* mutation status, the prospective nature of our registry, and the long follow-up period. We did not include information regarding neoadjuvant or adjuvant oncological treatment in this study. As it is widely known, BC treatment has evolved exponentially during the last two decades. An integrative analysis with neoadjuvant or adjuvant chemotherapy, as well as treatment with HER-2 targeted therapy and/or hormonal therapy, would probably give us relevant data regarding iDFS and OS, particularly in patients treated during the past ten years. Furthermore, missing data for some patients was also a limitation of this study. Our registry is centralized in one center but includes patients referred for genetic testing from several hospitals around the country. Although our cohort included patients who consented to follow-up, allowing registry updates, the management of primary tumors and risk-reducing surgeries may be performed either at our center or in other institutions. This may have resulted in underreporting of recurrence, subsequent tumors, and risk-reducing surgeries.

We hope our study will be useful in expanding knowledge, further clarifying differences in prognosis for *BRCA1-* and *BRAC2*-associated BC, and helping to improve counseling, risk management, and treatment strategies.

## 5. Conclusions

The presence of *BRCA1/2* pathogenic variants confers a higher risk of relapse and death in patients with BC. Among patients with these variants, although no statistical difference was observed, *BRCA2* BC patients tended to have better prognosis in iDFS and OS than *BRCA1*. This may be related to the early onset and predominant triple-negative phenotype of the latter, compared with older age and mostly hormone receptor-positive phenotype in *BRCA2* BC patients. Risk-reducing surgeries, namely contralateral prophylactic mastectomy and preventive salpingo-oophorectomy, confer a significantly lower risk of relapse and longer iDFS and OS, respectively. Therefore, both surgeries should be discussed during clinical management. This discussion should take into consideration BC stage, risk of relapse, and potential impact on relapse and survival.

To the best of our knowledge, this is the largest prospective study regarding the Portuguese *BRCA1/2* population and contains the highest number of *BRCA2* patients.

## Figures and Tables

**Figure 1 cancers-15-05699-f001:**
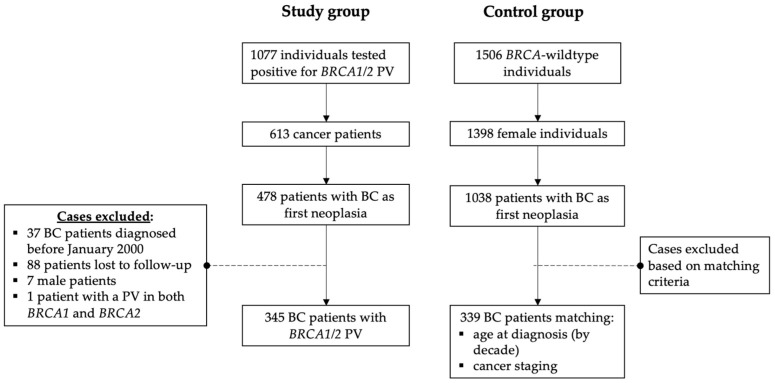
Flowchart of included patients. BC, breast cancer; PV, pathogenic variant.

**Figure 2 cancers-15-05699-f002:**
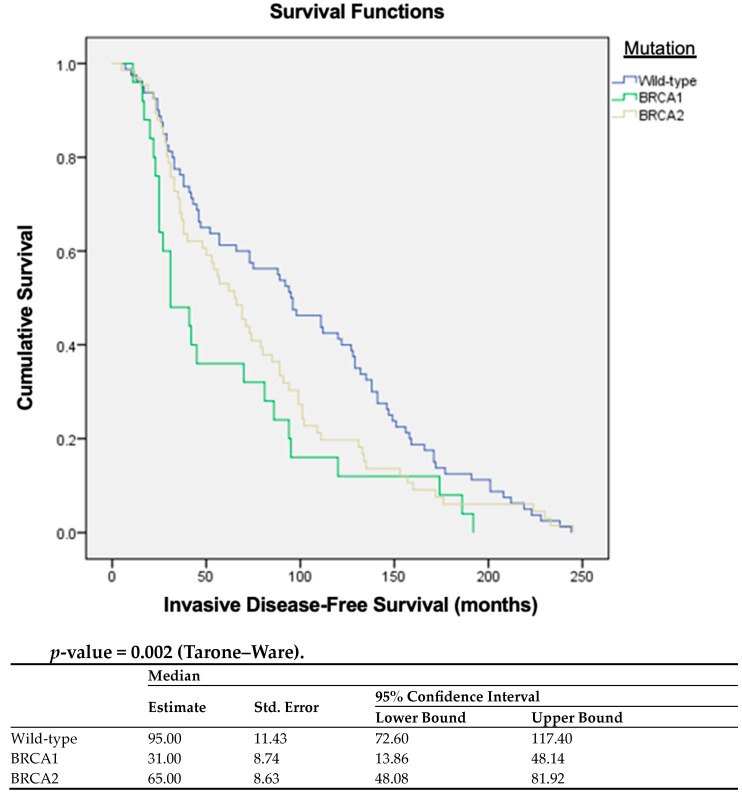
Kaplan–Meier curves with estimates of iDFS comparing patients with *BRCA*-wt, *BRCA1*, and *BRCA2* mutation.

**Figure 3 cancers-15-05699-f003:**
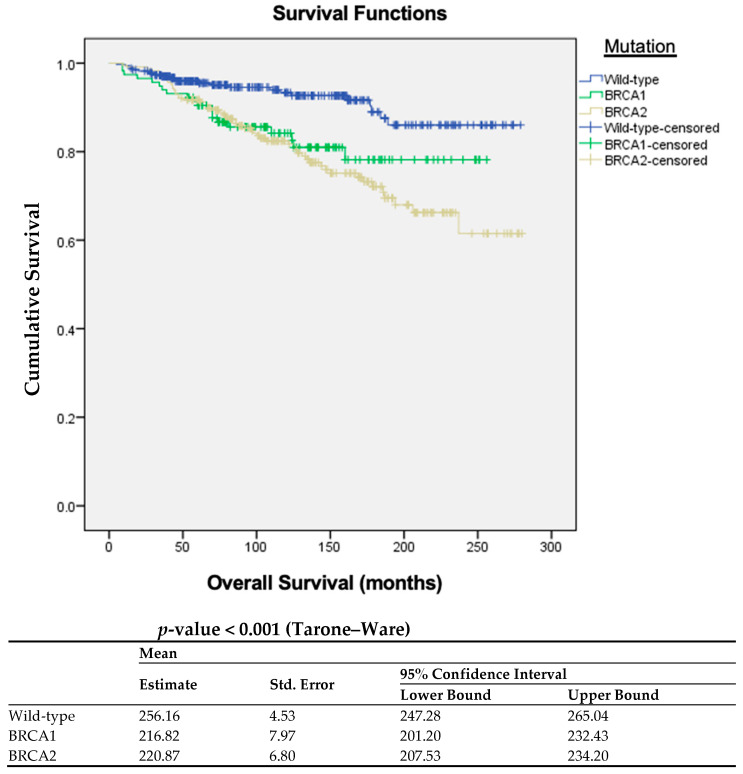
Kaplan–Meier curves with estimates of OS comparing patients with *BRCA*-wt status, *BRCA1* mutation, and *BRCA2* mutation.

**Table 1 cancers-15-05699-t001:** Patients’ characteristics.

	*BRCA*-wt N = 339	*BRCA1* N = 116	*BRCA2* N = 229	*p*-Value
Age at diagnosis—median (IQR)	44.0 (21–88)	40.5 (28–64)	42.0 (25–80)	**0.020**
Molecular subtype—n (%)				**<0.001**
Luminal A	221 (65.2)	30 (26.1)	170 (75.6)	
Luminal B	40 (11.8)	6 (5.2)	17 (7.6)	
HER2 enriched	21 (6.2)	3 (2.6)	4 (1.8)	
Triple-negative	57 (16.8)	76 (66.1)	34 (15.1)	
Clinical stage IV at diagnosis—n (%)	6 (1.8)	4 (3.5)	6 (2.7)	0.357
Relapse (yes)—n (%)	79 (23.7)	25 (21.9)	66 (29.3)	0.203
Vital status—n (%)				**<0.001**
Alive	315 (92.9)	96 (82.8)	174 (76.0)	
Dead	24 (7.1)	20 (17.2)	55 (24.0)	

IQR, interquartile range. Statistical significance was set at *p* = 0.05 (two-sided) in bold.

**Table 2 cancers-15-05699-t002:** *BRCA1* and *BRCA2* patients and uptake of prophylactic mastectomy.

Patients Undergoing Mastectomy	No	Yes	*p*-Value
Invasive disease (yes)—n (%)	28 (33.7)	17 (10.2)	**<0.001**
iDFS—median (IQR)	40.0 (83)	70.0 (60)	0.543
OS—median (IQR)	127.0 (111)	119.0 (92)	0.898

IQR, interquartile range; iDFS, invasive disease-free survival; OS, overall survival. Statistical significance was set at *p* = 0.05 (two-sided) in bold.

**Table 3 cancers-15-05699-t003:** *BRCA1* and *BRCA2* patients and uptake of salpingo-oophorectomy.

Patients Undergoing Salpingo-Oophorectomy	No	Yes	*p*-Value
Invasive disease (yes)—n (%)	23 (34.3)	44 (65.7)	**0.012**
iDFS—median (IQR)	33.0 (39)	73.5 (90)	**0.001**
OS—median (IQR)	105.5 (110)	135.0 (91)	**0.015**

IQR, interquartile range; iDFS, invasive disease-free survival; OS, overall survival. Statistical significance was set at *p* = 0.05 (two-sided) in bold.

**Table 4 cancers-15-05699-t004:** Second primary cancer types diagnosed during follow-up according to *BRCA* status.

	*BRCA*-wt	*BRCA1*	*BRCA2*
N (%)	20 (5.9)	22 (19.0)	48 (21.0)
Bladder	0	1	0
Breast	2	9	27
CNS	0	0	1
Colorectal	3	1	0
Endometrial	5	1	0
Gastric	0	0	3
Head and neck	0	1	1
Kidney	0	0	1
Lung	1	0	2
Lymphoma/leukemia	2	1	2
Ovary	2	6	7
Pancreatic	2	1	2
Sarcoma	2	0	0
Thyroid	1	1	2

**Table 5 cancers-15-05699-t005:** Predictors of OS and iDFS on multivariate analyses.

Subgroup/Predictor(s)				
	B	Exp (B)	95% CI for Exp (B)	*p*
OS				
Mastectomy	−1.624	0.197	0.082–0.475	**0.000**
iDFS				
Oophorectomy	−0.899	0.407	0.215–0.773	**0.006**

Statistical significance was set at *p* = 0.05 (two-sided) in bold.

## Data Availability

The data, in aggregate form, that support the findings of this study are available on request from the corresponding author. The data are not publicly available due to privacy or ethical restrictions.

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
