# Peer review of "Comparing Prognosis for BRCA1, BRCA2, and Non-BRCA Breast Cancer"

_cancers, 2023, doi:10.3390/cancers15235699_

Round 1
Reviewer 1 Report
Comments and Suggestions for Authors
Meireles et al have submitted an interesting retrospective study comparing the prognosis of BRCA1, BRCA2 and BRCA-wt breast cancer among 684 patients. Overall, the manuscript is well-organized and written. Authors have shown very interesting data for both genes BRCA1/2. However, there are very minor comments that might be taken into consideration to improve the manuscript.
1- The aim of the paper is missing in abstract.
2- For the IQR, it’d be better to include both the lower and upper half of the data.
Comments on the Quality of English LanguageMinor editing of English language required.
Reviewer 2 Report
Comments and Suggestions for Authors
I would like to thank the authors for this interesting article. They report the results of a prospective follow-up registry of BRCA 1 and BRCA2 patients with a mean follow-up time of 113.0 months. Overall survival, as well as invasive disease free survival are compared between BRCA patients and BRCA-wt. All in all, it is a interesting topic and the article is well written and easy to follow.
Please find enclosed some suggestions for improvement:
1. Your mean follow-up time is 113.0 months. What was the standard deviation? What was the maximum and minimum follow-up time? I think it would be nice to include this information in the text. Furthermore, if the follow-up time is not normally distributed it would be more accurate to use the the median (with minimum/maximum) rather than the mean (with standard deviation).
2. Materials and methods: Line 80: “If considered eligible for BRCA1/2 testing” – can you please include the eligibility criteria of Portugal as this might be different compared to other countries?
3. Materials and Methods, Line 86-88: “ A control group, 1:1, matched for age and stage , was identified… “ – Can you please exactly explain how this control group was identified or chosen? As all the results of the study rely on a comparison to that “control group” I think it might be very important to know exactly how this group was created. – Or which patients might have not been chosen for the control group…
4. IF patients in BRCAwt have been matched 1:1 – how is it possible that patients with BRCAwt were statistically significantly older than patients with BRCA 1 or BRCA 2.
5. Results: Patients characteristics: I don’t completely understand how the 684 patients of the study have been identified. You state that there were only 613 patients with cancer? Did not all the patients enrolled in this study have breast cancer? Were patients with other cancers also enrolled in this study?
Please explain this in more detail. A flowchart, where you can see exactly which patients were included or excluded (due to certain criteria) could help here.
6. Figure 1 and figure 2: What does “mutacao” on the right side of the figure mean? And what does “meses” at the bottom of the figure after “survival” mean? Please translate to English or erase. Furthermore, it would be nice to write “cumulative survival” not “cum survival”.
7. Discussion: Line 219: “multivariate analysis was performed”- I assume you did a Cox regression? – It would be nice to include those results in the results section as well.
Comments on the Quality of English LanguageIn my opinion the article is well written and was easy to follow. The materials and methods sections need to be more specified and written "more clearly".
Reviewer 3 Report
Comments and Suggestions for Authors
This is an interesting and well written summary of findings regarding prognosis of BRCA1/2 patients relative to BRCAwt patients with breast cancer. While it does not go so far as to suggest escalation of therapy for these patients' known cancer, it provides additional support for encouraging patients to undergo prophylactic mastecomy/oophorectomy.
Author Response
Dear Reviewer 3,
We would like to express our sincere gratitude for your invaluable time, dedication, and meticulous review of our manuscript. Your thoughtful feedback is greatly appreciated and has contributed to enhancing the quality of our work.
We are delighted that you have grasped the core messages of our work and consider it worthy of publication.
Best regards,
Pedro Antunes Meireles
Round 2
Reviewer 2 Report
Comments and Suggestions for Authors
All suggestions have been incorporated in the manuscript.